# Electrochemical $CO_2$ reduction to high-concentration pure formic acid solutions in an all-solid-state reactor

Lei Fan [1,2,5], Chuan Xia [1,3,5], Peng Zhu [1], Yingying Lu[2✉] & Haotian Wang [1,4✉]

Electrochemical $CO_2$ reduction reaction ($CO_2$RR) to liquid fuels is currently challenged by low product concentrations, as well as their mixture with traditional liquid electrolytes, such as $KHCO_3$ solution. Here we report an all-solid-state electrochemical $CO_2$RR system for continuous generation of high-purity and high-concentration formic acid vapors and solutions. The cathode and anode were separated by a porous solid electrolyte (PSE) layer, where electrochemically generated formate and proton were recombined to form molecular formic acid. The generated formic acid can be efficiently removed in the form of vapors via inert gas stream flowing through the PSE layer. Coupling with a high activity (formate partial current densities ~450 mA cm$^{-2}$), selectivity (maximal Faradaic efficiency ~97%), and stability (100 hours) grain boundary-enriched bismuth catalyst, we demonstrated ultra-high concentrations of pure formic acid solutions (up to nearly 100 wt.%) condensed from generated vapors via flexible tuning of the carrier gas stream.

---

[1] Department of Chemical and Biomolecular Engineering, Rice University, Houston, TX 77005, USA. [2] State Key Laboratory of Chemical Engineering, Institute of Pharmaceutical Engineering, College of Chemical and Biological Engineering, Zhejiang University, Hangzhou 310027, China. [3] Smalley-Curl Institute, Rice University, Houston, TX 77005, USA. [4] Azrieli Global Scholar, Canadian Institute for Advanced Research (CIFAR), Toronto, 22 Ontario M5G 1M1, Canada. [5] These authors contributed equally: Lei Fan, Chuan Xia. ✉email: yingyinglu@zju.edu.cn; htwang@rice.edu

Electrochemical carbon dioxide reduction reaction ($CO_2RR$) is changing the way we produce chemicals and fuels, while helping to mitigate climate change[1–8]. A variety of products ranging from hydrocarbons to oxygenates[9–17] and from $C_1$ to $C_3$ can be produced from $CO_2RR$ using different catalytic materials[18–24]. As the price of renewable electricity continues to decrease, the cost of some $CO_2RR$ products, particularly those single carbon molecules such as carbon monoxide (CO) and formate, becomes competitive to traditional chemical engineering processes due to their industrially relevant selectivity (>90%) and activity (>100 mA cm$^{-2}$)[25–36]. Compared with gas-phase $CO_2RR$ products, liquid products such as formate show significant advantages due to their high energy densities and ease of storage and distribution[37,38]. Currently, ca. 800,000 metric tons of formic acid (HCOOH) is produced globally every year for a wide range of applications, including chemical production, cleaning, textile industry, and antiseptics[39–41]. More importantly, with the fast development of dehydrogenation catalysts[42], formic acid now becomes an attractive hydrogen carrier due to its liquid phase under ambient conditions, high volumetric hydrogen density (53 g $H_2$ per liter of HCOOH), and low toxicity[43–45]. Producing this important energy carrier via electrochemical $CO_2RR$ will prevent net carbon emissions and make it a carbon neutral liquid fuel.

However, there has always been a challenge that retards the practical applications of $CO_2RR$ route for renewable formic acid synthesis: the generated formic acid product is typically in a mixture with liquid electrolyte and thus cannot be directly used without futher purification. The conventional liquid electrolyte used in $CO_2RR$ in either H-cell or flow-cell reactors, such as $KHCO_3$, $Na_2SO_4$, or KOH, mainly has two coupled functionalities: one is the ionic transportation between cathode and anode for efficient current flow, and the other is the collection of liquid products. Therefore, the obtained product from $CO_2RR$ is actually formate ions mixed with other impurity ions from solutes (such as $K^+$, $HCO_3^-$, etc.). This mixture necessities downstream separation steps to obtain pure formic acid or formic acid solutions, which however is not only energy- and cost-intensive, but also complicates the infrastructure for delocalized production[2,44,46]. To resolve this challenge, our strategy is to decouple the ionic conduction and product collection functionalities in the electrolyte. We propose to utilize a porous solid electrolyte (PSE) layer, instead of traditional liquid electrolytes, between the cathode and anode for fast ionic transportations, where electrochemically generated formate and proton are recombined to form pure formic acid molecule[46]. As the solid electrolyte is insoluble, those formic acid molecules can then be efficiently collected in the deionized (DI) water flow through the PSE layer, without involving any impurity ions. Although our recent research has demonstrated the feasibility of solid electrolyte design for obtaining pure formic acid solutions[46], a few challenges still exist that impede its practical applications: the product concentration was limited due to the DI water flow stream in the solid electrolyte layer where a significant amount of water was present; the product generation rate was not sufficiently high for industrial applications; and it still involves the use of liquid electrolyte on the anode side for water oxidation.

Herein, we report electrochemical $CO_2$ reduction to formic acid vapors using an all-solid-state reactor. The formic acid vapor product can either be easily condensed as high-concentration liquid fuel or directly fed into a gas phase reactor for its following applications. Our design can avoid the challenges caused by the usage of liquid electrolytes and can generate high-purity and high-concentration formic acid. Besides, all-solid-state reactors are suitable for modular and high-pressure systems in future large-scale deployments. As shown by the schematic in Fig. 1a, the cathode and anode were separated by a PSE layer. On the cathode side, $CO_2$ can be selectively reduced to formate ions on a selective $CO_2RR$ catalyst; on the anode side, instead of using liquid acid for the oxidation reaction that could complicate the device level engineering, here we fed in a $H_2$ gas stream for the hydrogen oxidation reaction (HOR) to release protons. Gas diffusion layer (GDL) electrodes coated with $CO_2RR$ and HOR catalysts were used as cathode and anode to improve the mass transfer of both $CO_2$ and $H_2$. Besides, both catalysts were in close contact with anion and cation exchange membranes (AEM and CEM), respectively, for efficient ion transportation. Electrochemically generated $HCOO^-$ and $H^+$ were driven by the electric field to move across the AEM or CEM into the PSE layer, and were recombined to form HCOOH molecule. Instead of using a DI water stream to discharge the product, which would significantly bring down the formic acid concentration, here we report the use of an inert gas ($N_2$) flow to carry away formic acid vapors for high-concentration products. When using a DI water flow stream to get high concentration of pure formic acid solutions, the flow rate should be lowered, while maintaining the product generation rate. As the formic acid concentration within the solid electrolyte layer gets higher and higher, the Faradaic efficiency (FE) is dramatically decreased due to the potentially increased reverse reaction rate, as well as the increased crossover to the anode as frequently observed in fuel cells[47,48]. However, by using $N_2$ gas flow to bring out the formic acid vapors, the gas flow rate can be kept high so as to avoid the accumulation of products within the solid electrolyte layer. High concentration of formic acid can still be obtained by a simple cold-condensation process, while the $N_2$ stream can go back to the reactor. As a result, we can achieve concentrated products while not sacrificing much selectivity compared with the DI water flow design. Coupling with a high-activity (formate partial current densities > 440 mA cm$^{-2}$), high-selectivity (maximal FE > 97%), and high-stability (100 h) grain boundary (GB)-enriched bismuth (Bi) catalyst, we demonstrated high concentrations of pure formic acid solutions (up to nearly 100 wt.%) condensed from its vapor by tuning the gas flow stream through the PSE layer, representing a superior catalytic performance compared with the existing systems (Fig. 1b and Supplementary Table 1).

## Results and discussion
**Fabrication and characterizations of nBuLi-Bi catalyst.** We selected Bi as the catalytic material for $CO_2RR$ due to its intrinsic high-selectivity towards formate[35,36,49,50]. However, the catalytic performance of existing Bi catalysts still needs to be improved, particularly in its formate partial currents, for practical applications. Electronic structure and morphology tuning methods, such as metal alloying, two-dimensional layered materials, quantum-dot-derived catalysts, and so on, have been demonstrated to improve the $CO_2RR$ performance of different catalysts[30,51–54]. Among them, metal oxide-derived materials, with GBs formed during the electrochemical reduction process, have aroused particular interests due to their significantly improved $CO_2RR$ selectivity and activity[55–58]. However, for some low-conductivity metal oxides, such as $Bi_2O_3$ and ZnO, it is difficult to form GBs by electrochemical reduction methods due to its slow process[59]. To further boost Bi catalysts' $CO_2RR$ activity, here we report a chemical lithium (Li) tuning method to create abundant active GBs. Commercially available $Bi_2O_3$ micro powders were selected as a preparation for potential large-scale uses in future's practice. As illustrated in Fig. 2a, $Bi_2O_3$ was first soaked into n-butyl lithium (nBuLi) and react at 80 °C for 24 h. During this process, Li ions were chemically intercalated into $Bi_2O_3$ lattices, reducing $Bi_2O_3$ into ultra-small metallic Bi nanoparticles embedded in the $Li_2O$ matrix. The treated Bi-Li-O powder was then soaked in

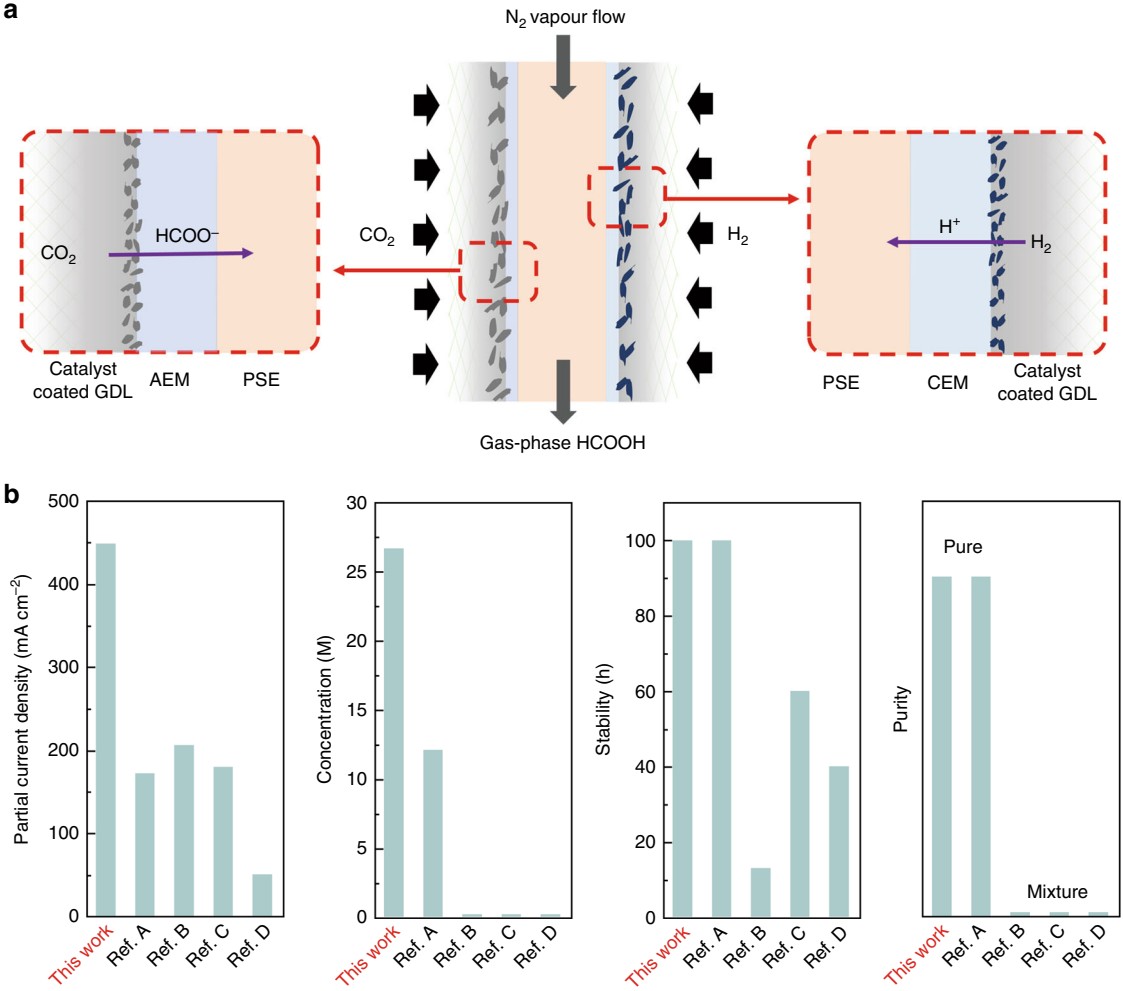

**Fig. 1 Schematic of all-solid-state electrochemical CO₂RR reactor and performance comparisons. a** Schematic illustration of the all-solid-state electrochemical CO₂RR to formic acid reactor. AEM anion exchange membrane, CEM cation exchange membrane, GDL gas diffusion layer, PSE porous solid electrolyte. Electrochemically generated HCOO⁻ and H⁺ ions were driven by the electric field to move across the AEM or CEM into the PSE layer and recombined to form formic acid molecule, which can be released quickly in its vapor phase via the N₂ carrier gas flow. **b** The electrochemical performance of our all-solid-state CO₂RR reactor compared with previous literature (A: ref. [46], B: ref. [36], C: ref. [35], D: ref. [31]).

water to violently leach out all Li₂O and remaining Li compounds, forming ultra-small Bi nanoparticles with GBs created. Scanning electron microscopy (SEM) was employed to observe the morphology change of the Bi catalysts before and after nBuLi treatment. As shown in Fig. 2b, the particle size of pristine Bi₂O₃ was from hundreds of nanometers to a few micrometers. After nBuLi tuning, the particle size was dramatically decreased to only tens of nanometers, providing more active sites for CO₂RR (Fig. 2c). This was further confirmed by the Brunauer–Emmett–Teller (BET) measurement that the surface area of nBuLi-treated Bi (nBuLi-Bi) was 9.5 times higher than the commercial Bi₂O₃ (Supplementary Fig. 1). The increased surface area could boost the contact between the catalyst and membrane in membrane electrode assemblies, leading to enhanced CO₂RR performance. X-ray diffraction (XRD) measurement suggested that, after the nBuLi treatment, the Bi₂O₃ precursor has been successfully converted into metallic Bi (Supplementary Fig. 2), which is the active phase under CO₂RR conditions[46]. The result is consistent with the X-ray photoelectron spectroscopy (XPS) analysis (Supplementary Fig. 3). This prefabricated metallic Bi ultra-small nanoparticles help to avoid the possible particle aggregation during the oxide prereduction process, maintaining those generated active GBs. The high-resolution transmission

electron microscopy (TEM) image of Bi₂O₃ precursor in Fig. 2d showed its single crystalline nature. After the Li treatment, small-size grains with distinguished lattice orientation was observed (Fig. 2e), suggesting the formation of GBs.

**Electrocatalytic CO₂RR performance of nBuLi-Bi catalyst**. The intrinsic CO₂RR-to-formate performance on nBuLi-Bi was first evaluated in a standard three-electrode flow cell, with 1.0 M KHCO₃ solution used as the neutral electrolyte ("Methods"). The catalyst was uniformly air-brushed onto a GDL electrode with a mass loading about 0.7 mg cm⁻². The thickness of the catalyst layer was measured to be ~55 μm (Supplementary Fig. 4a). Compared with the layered structure we used in our previous study where the layer-by-layer stacking could block the CO₂ diffusion pathway, the porous nanoparticulate morphology of nBuLi-Bi (Supplementary Fig. 4b) now allows for more efficient CO₂ diffusions from GDL to the surface of the catalyst layer, which, as a result, promotes higher current densities while maintaining good formate FEs[60]. Different from traditional flow-cell reactor where catalysts were typically in a direct contact with the liquid electrolyte, in our flow cell, an AEM was used in the cathode side to separate the catalysts with liquids and thus avoid

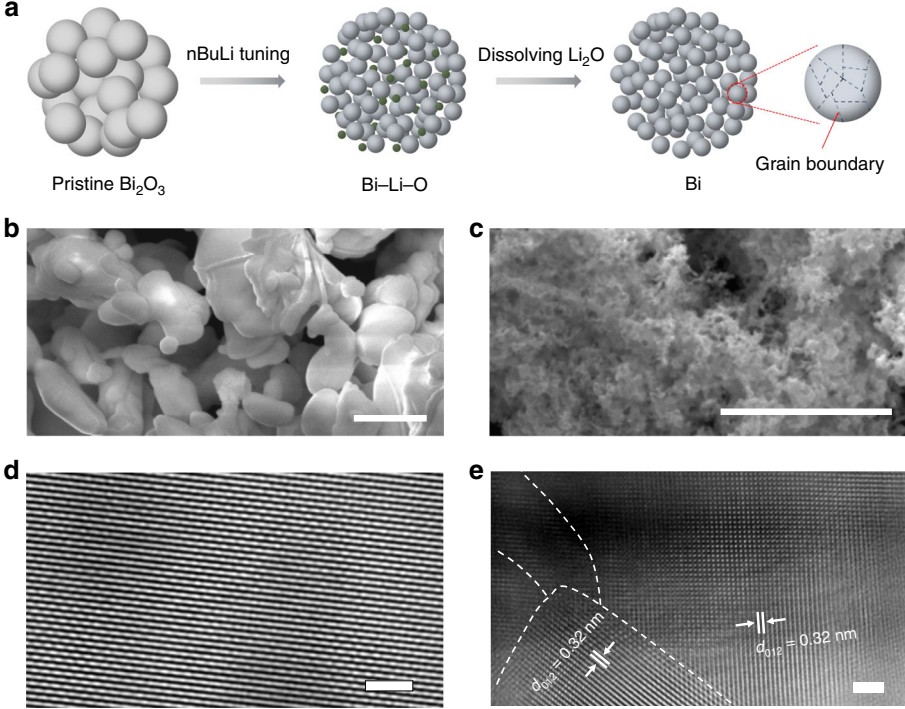

**Fig. 2 Characterization of Bi$_2$O$_3$ precursor and nBuLi-Bi catalyst. a** Schematic illustration of the fabrication process of nBuLi-Bi catalyst. **b**, **c** SEM images of pristine Bi$_2$O$_3$ and nBuLi-Bi, respectively. **d**, **e** HRTEM image of pristine Bi$_2$O$_3$ and nBuLi-Bi, respectively. Different orientations of grains were observed in nBuLi-Bi in **e**. Scale bars: **b**, **c** 1 μm; **d**, **e** 2 nm.

the flooding issues for good stability. Gas and liquid products were detected by gas chromatograph (GC) and nuclear magnetic resonance (NMR), respectively ("Methods"). Due to the smaller particle size and enriched GBs, nBuLi-Bi achieved 5 mA cm$^{-2}$ reduction current at only −0.67 V vs. reversible hydrogen electrode (RHE), while the electrochemically reduced Bi from Bi$_2$O$_3$ precursor (OD-Bi) needs −0.72 V vs. RHE to deliver the same amount of current (Fig. 3a). With the overpotential gradually increased, the formate FE of nBuLi-Bi ramped up to a maximal of 97% (Fig. 3b), with small amount (<4%) of H$_2$ and carbon monoxide side products detected (Supplementary Fig. 5). Over 90% formate selectivity was maintained over a wide electrochemical window from −0.72 to −1.05 V, in a sharp contrast with that of OD-Bi. At −1.05 V, nBuLi-Bi reached an impressive overall current density of up to 500 mA cm$^{-2}$, while still maintaining a high formate selectivity of 92%, achieving a formate partial current of 460 mA cm$^{-2}$ in neutral pH. This represents a nearly fourfold improvement compared with that of OD-Bi without Li treatment (Fig. 3c). The higher activity of nBuLi-Bi could be ascribed to the in situ generated GBs, as well as the significantly increased surface area.

To confirm whether the high intrinsic formate activity of nBuLi-Bi can be successfully translated in our solid electrolyte reactor for pure formic acid solutions, we first use DI water flow stream through the PSE layer for product collection ("Methods")[46,61,62]. The DI water stream was set at 54 mL min$^{-1}$ in a 4.75 cm$^2$ cell to maintain a fast-enough rate, which avoids significant product accumulations within the PSE layer even under high production rate for intrinsic selectivity evaluation. As for the anode, commercial platinum on carbon (Pt/C) as the state-of-the-art HOR catalyst was selected for low overpotentials and high conversion rate[63–66]. We employed the styrenedivinyl-benzene copolymer microspheres with sulfonic acid functional groups as the PSE, which enables fast proton transportations under room temperature from the anode to the cathode[67]. As

shown in Fig. 3d, where pure formic acid solutions were produced (Supplementary Fig. 6), an onset cell voltage was observed at ~1.28 V to deliver a current of 8.4 mA cm$^{-2}$. With higher cell voltage applied, the cell current was rapidly increased to a few hundreds of mA cm$^{-2}$, while still maintaining a high formic acid FE of above 80% until a large cell current density of 421 mA cm$^{-2}$ (Fig. 3e). With even higher currents delivered (up to 842 mA cm$^{-2}$ or 4 A of cell current), the formic acid selectivity started to show an obvious decrease possibly due to the CO$_2$ mass transport limitation as well as the correspondingly increased HER side reaction. At a cell voltage of 2.19 V, a maximal formic acid partial current density of 440 mA cm$^{-2}$ was achieved, corresponding to the generation of 0.62 M pure formic acid solution at a rate of 8.21 mmol h$^{-1}$ cm$^{-2}$ (Fig. 3e and Supplementary Fig. 7). In comparison, the OD-Bi required much higher cell voltage to deliver the same current density in the PSE cell. The maximum formic acid partial current density was only 174.4 mA cm$^{-2}$ at a cell voltage of 2.44 V (Supplementary Fig. 8). More importantly, we showed that the concentration of pure formic acid solutions can be easily tuned by controlling the DI water flow rate. By gradually slowing down the DI water flow rate while maintaining a cell current density of 420 mA cm$^{-2}$, the formic acid concentration gradually increased to a maximal of 2.85 M (~12.8 wt.%, Supplementary Fig. 9). Possible impurities in collected products, examined by inductively coupled plasma atomic emission spectroscopy (ICP-OES), such as sodium, potassium (common impurity ions in water), iron (from device), sulfur (from solid electrolyte), bismuth (from cathode), and platinum (from anode), were at p.p.m. or lower level (Supplementary Table 2), demonstrating the ultra-high purity of our generated formic acid solutions. Therefore, those electrochemically synthesized pure formic acid solutions are ready for immediate use without any further purification processes, which is economically promising (Supplementary Note 1). We also demonstrate the stability of our nBuLi-Bi catalyst in the PSE cell

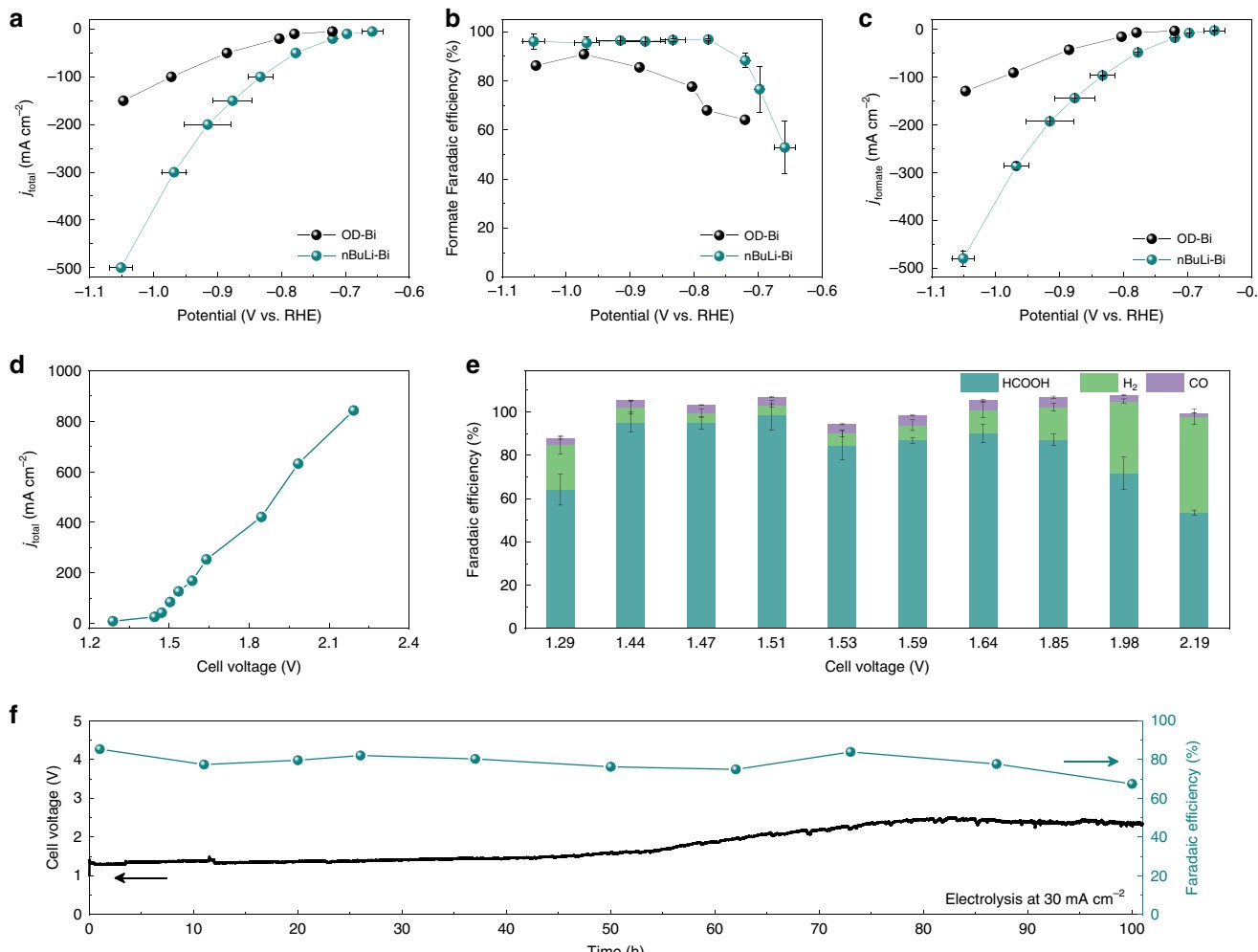

**Fig. 3 Electrochemical CO₂ reduction performance of OD-Bi and nBuLi-Bi. a** The *I–V* curves of OD-Bi and nBuLi-Bi catalysts in a flow-cell setup using 1.0 M KHCO₃ electrolyte. **b** The corresponding formate Faradaic efficiencies (FEs) under different potentials. The maximal formate FE of nBuLi-Bi is 97% at −0.77 V vs. RHE. **c** The comparison of formate partial current between OD-Bi and nBuLi-Bi. The maximal formate partial current on nBuLi-Bi reached to 450 mA cm⁻², which represents a fourfold improvement from OD-Bi. **d** The *I–V* curve of PSE cell with nBuLi-Bi and Pt/C as the CO₂RR and HOR catalysts, respectively, where DI water was used to release the produced formic acid molecules. **e** The corresponding FEs of different products under different cell voltages. **f** Stability test of continuous generation of pure formic acid solutions with concentrations over 0.1 M. The cell current and DI water flow rate was fixed at 30 mA cm⁻² and 16 mL h⁻¹, respectively. The full cell voltage showed a slight increase after 50 h, which could stem from the anode/CEM interface. The error bars represent two independent tests.

using DI water flow ("Methods"). A 100 h continuous and stable production of ~0.1 M pure formic acid solution was achieved (Fig. 3f). The current density was fixed at 30 mA cm⁻² (142.5 mA cell current) with a DI water flow rate of 16 mL h⁻¹. Although the full cell voltage showed a slight increase after 50 h operation, which is possibly related with the weak Pt/C-CEM contact as observed after catalysis, the formic acid selectivity remained relatively stable during the full course of stability test. Long-term stability at high current density of 100 mA cm⁻² was also demonstrated (Supplementary Fig. 10). After 10 h continuous operation, the formic acid FE was still over 80%. The quick degradation of CO₂RR stability at high current density could be mainly caused by the degradation of AEM, as the AEM could be partially neutralized by CO₂ gas[68]. As a result, the ionic conductivity of the AEM will gradually decrease and thus leads to increased cell voltage. When the current density increased, higher concentration of hydroxide ions will be generated simultaneously during CO₂RR[69]. Then, the CO₂ permeation flux will increase, leading to faster CO₂RR performance decay.

**Production of pure formic acid vapor**. As discussed above, high-concentration pure formic acid solutions could be obtained by slowing down the flow rate of DI water. However, as the formic acid concentration within the solid electrolyte layer got higher and higher, the FE was dramatically decreased (Supplementary Fig. 11), due to the potentially increased reversible reaction rate, as well as the increased crossover to the anode as frequently observed in fuel cells[47,48]. The highest formic acid concentration we obtained from DI water flow is up to ~8.3 M by slowing the DI water flow rate down to as low as 0.5 mL h⁻¹, whereas the product selectivity dropped to only ~20% (Supplementary Fig. 11). To further improve the formic acid concentrations while maintaining high product selectivity, our strategy is to use N₂ gas flow instead of the DI water flow to carry out generated formic acid molecules ("Methods"). In the case of DI water steam, the formic acid concentration within the PSE layer is the same with collected products; when an inert gas stream such as N₂ is used instead, those generated formic acid molecules can be efficiently carried out as vapor phase, keeping a relatively low formic acid

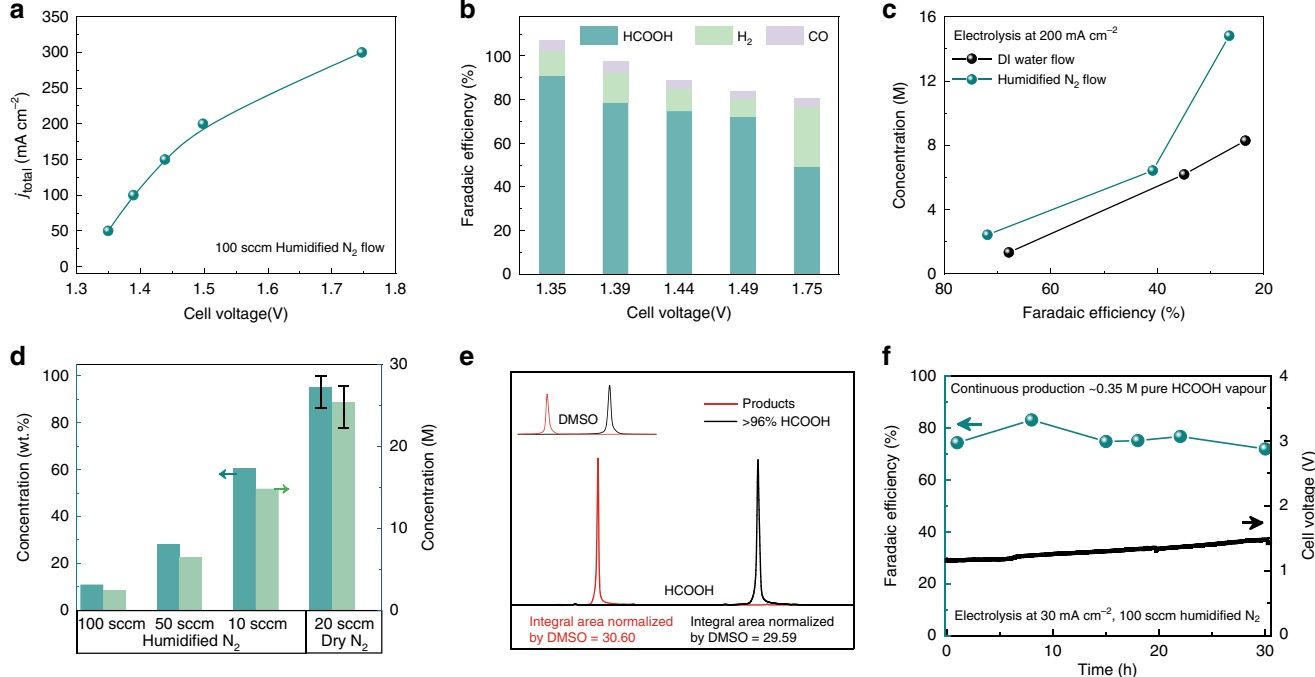

**Fig. 4 All-solid-state electrochemical CO₂RR reactor for high-concentration formic acid vapor. a** The *I–V* curve of all-solid-state reactor using 100 s.c.c. m. humidified $N_2$. **b** The corresponding FEs of formic acid vapor under different cell voltages. **c** Performance comparison between DI water flow and humidified $N_2$ flow. Due to the high formic acid vapor release rate, humidified $N_2$ flow can maintain a high product selectivity, while achieving concentrated formic acids. **d** The dependence of formic acid product concentration on the $N_2$ gas flow rate under a fixed overall current density of 200 mA cm⁻². By utilizing dry $N_2$ gas flow, up to nearly 100 wt.% pure formic acid was achieved. **e** The NMR spectrum of formic acid product obtained via dry $N_2$ flow, compared with that of commercial formic acid product with a marked purity of >96 wt.%. The normalized integral area of our formic acid products was even slightly higher than commercial product. **f** Stability test of CO₂RR to formic acid vapor under a current density of 30 mA cm⁻². The error bars represent three independent tests.

concentration within the PSE layer to maintain a high selectivity (Supplementary Fig. 12), while still obtaining high-concentration products by a simple downstream cold-condensation process.

We first validated that formic acid vapor can be successfully brought out by humidified $N_2$ gas flow as shown in the *I–V* curve in Fig. 4a, with the corresponding FEs presented in Fig. 4b. Up to 300 mA cm⁻² current density was obtained in this all-solid-state cell. As suggested in Fig. 4c, although there is not a big difference in product selectivity under low concentration region, the advantage of inert gas flow stands out in producing more concentrated products than DI water flow under the same product selectivity. Under the fixed humidified $N_2$ flow rate of 100 standard cubic centimeters per minute (s.c.c.m.) and overall current of 200 mA cm⁻² in our 4.75 cm² all-solid-state cell, the condensed formic acid solution from the vapor was up to 11.0 wt. % or 2.4 M (Supplementary Fig. 13). Importantly, the formic acid selectivity was still maintained at ~72% under such high product concentration (Fig. 4b), showing a dramatic improvement compared with the DI water flow under similar reaction conditions (Fig. 4c). When we further lower the humidified $N_2$ flow rate to 10 s.c.c.m., the condensed formic acid concentration was increased to 60.5 wt.% or 14.8 M (Fig. 4d), which has never been achieved by using DI water flow[46,62]. To further push the product concentration limit, we used a 20 s.c.c.m. dry $N_2$ flow to carry out formic acid vapor with minimized water vapor involvement. Surprisingly, we achieved nearly 100 wt.% pure formic acid product in the downstream cold trap. As shown in the NMR spectra comparison in Fig. 4e and Supplementary Fig. 14, our formic acid product presented even higher purity compared with that of commercial formic acid standard with a marked purity of >96 wt.% ("Methods"), suggesting the

generation of ultra-pure formic acid from our all-solid-state reactor. To our best knowledge, this is the first time that pure formic acid can be directly obtained via electrochemical synthesis without any downstream separation or purification processes. We noted that the dry $N_2$ flow for ultra-pure product may not be suitable at current state for practical operations due to the gradual degradation of membranes and solid electrolytes without wetting conditions (Supplementary Figs. 15 and 16, and Supplementary Note 2); however, this exciting result demonstrates that our all-solid-state reactor design can in principle produce ultra-pure liquid fuels via CO₂RR electrosynthesis with no need of product separation and purification processes. The performance can be further improved in future's research via solid electrolyte engineering, membrane improvement under dry conditions, operations under elevated temperature, etc. With humidified $N_2$ gas flow, our all-solid-state cell demonstrated an impressive stability of 30 h continuous production of ~0.35 M pure formic acid product (Fig. 4f).

In summary, we demonstrated a strategy to directly synthesize gas-phase formic acid via electrocatalytic CO₂RR in an all-solid-state reactor. A highly active and stable Bi catalyst, with rich GBs induced by Li ions, was employed to efficiently drive the CO₂-to-HCOOH conversion (>95% FE) under significant rates. Different from traditional liquid electrolytes where generated liquid fuels were mixed, our PSE layer helps with fast ion conduction, while not introducing any impurity ions. More importantly, those generated formic acid molecules within the PSE layer can be continuously and stably carried out in vapor form by $N_2$ gas flow, resulting in unprecedented product purities and concentrations compared with existing systems. Our design also provides a potential route to store gas-phase $H_2$ into liquid phase HCOOH.

Our all-solid-state electrochemical system could be further extended to other electrocatalytic reactions for high-purity and high-concentration products. Future research can be focused on improving the performance of catalysts in both cathode and anode, the stability of solid electrolyte, and ion exchange membranes under low water conditions, preventing product crossover.

## Methods

**Preparation of nBuLi-Bi.** $Bi_2O_3$ (300 mg, Sigma Aldrich) was mixed with 10 mL nBuLi (2.5 M in hexanes, Sigma Aldrich) in an argon-filled glove box. The above solution was heated to 80 °C for 24 h under continuous stirring. After cooling to room temperature, the obtained products were washed using excess dry hexane to remove any possible residual nBuLi. The treated powder was then soaked in water to violently leach out all $Li_2O$ and remaining Li compounds. The powder was further washed using DI water and isopropanol, and then collected by centrifugation. Finally, the as-obtained catalyst was dried at 60 °C.

**Characterization of the catalysts.** SEM was performed on a FEI Quanta 400 field-emission SEM. TEM characterization was carried out using a FEI Titan Themis aberration-corrected TEM at 300 kV. Powder XRD data were collected using a Bruker D2 Phaser diffractometer in parallel beam geometry employing Cu Kα radiation ($\lambda$ = 1.54056 Å) and a one-dimensional LYNXEYE detector, at a scan speed of 0.02° per step and a holding time of 1 s per step. XPS was obtained with a PHI Quantera spectrometer, using a monochromatic Al Kα radiation (1486.6 eV) and a low-energy flood gun as neutralizer. All XPS spectra were calibrated by shifting the detected carbon C 1 s peak to 284.6 eV. BET surface area analysis was performed using Quantachrome Autosorb-iQ-MP/Kr BET Surface Analyzer. ICP-OES results were tested by a third-party company (A&B Lab) at Houston, TX.

**Electrochemical $CO_2$ reduction.** All the electrochemical measurements were run at 25 °C. A BioLogic VMP3 workstation was employed to record the electrochemical response. The typical three-electrode measurements were performed using a conventional flow cell. For aqueous electrolyte test, around 0.7 mg cm$^{-2}$ nBuLi-Bi catalyst was loaded on Sigracet 35 BC GDL electrode (Fuel Cell Store) as the cathode. Around 0.7 mg cm$^{-2}$ $IrO_2$-C catalyst (Fuel Cell Store) was loaded on Sigracet 35 BC GDL electrode as the anode for water oxidation. The two electrodes were therefore placed on opposite sides of two 0.5 cm-thick Polytetrafluoroethylene (PTFE) sheets with 0.5 cm-wide by 2.0 cm-long channels such that the catalyst layer interfaced with the flowing liquid electrolyte. A Nafion 115 film (Fuel Cell Store) was sandwiched by the two PTFE sheets to separate the chambers. An AEM (Dioxide Materials and Membranes International, Inc.) was added between the liquid electrolyte and the bismuth catalyst to avoid flooding. The geometric surface area of catalyst is 1 cm$^2$. On the cathode side, a titanium gas flow chamber supplied 50 s.c.c.m. (monitored by Alicat Scientific mass flow controller) humidified $CO_2$. In addition, the anode and cathode were circulated with 1 M $KHCO_3$ under a flow rate of 2 mL min$^{-1}$ for $CO_2$ reduction. A saturated calomel electrode (SCE, CH Instruments) was used as the reference electrode. All potentials measured against SCE was converted to the RHE scale in this work using $E_{RHE} = E_{SCE} + 0.244$ V + $0.0591 \times$ pH, where pH values of electrolytes were determined by Orion 320 PerpHecT LogR Meter (Thermo Scientific). Solution resistance (Rs) was determined by potentiostatic electrochemical impedance spectroscopy at frequencies ranging from 0.1 Hz to 200 kHz. All the measured potentials using three-electrode setup were manually compensated.

For the proposed two-electrode all-solid-state cells for pure HCOOH vapor production, an AEM (Dioxide Materials and Membranes International, Inc.) and a Nafion film (Fuel Cell Store) were used for anion and cation exchange, respectively. Around 0.7 mg cm$^{-2}$ nBuLi-Bi and Pt-C loaded on Sigracet 35 BC GDL electrode (4.75 cm$^2$ electrode area) were used as a cathode and anode, respectively. The cathode side was supplied with 50 s.c.c.m. of humidified $CO_2$ gas. The anode side was supplied with 50 s.c.c.m. of humidified $H_2$ gas. The porous styrenedivinylbenzene sulfonated copolymer were used as solid ion conductors. Humidified $N_2$ or DI water were used to release the produced HCOOH in the solid-state electrolytes. Specifically, a 250 mL glass bottle containing ~100 mL DI water was used for humidification, the glass bottle was heated in a hot plate, and the water temperature in the glass vial was ~80 °C. For pure HCOOH vapor (~100 wt.%) synthesis, dry $CO_2$, $H_2$, and $N_2$ were used to avoid the introduction of any water. All the measured potentials using two-electrode setup were manually compensated.

**$CO_2$ reduction product analysis.** To quantify the gas products during electrolysis, $CO_2$ gas (Airgas, 99.995%) was delivered into the cathodic compartment at a rate of 50.0 s.c.c.m. and vented into a GC (Shimadzu GC-2014) equipped with a combination of molecular sieve 5A, Hayesep Q, Hayesep T, and Hayesep N columns. A thermal conductivity detector was mainly used to quantify $H_2$ concentration and a flame ionization detector with a methanizer was used for quantitative analysis of CO content and/or any other alkane species. The partial current density for a given product was calculated as below:

$$j_i = x_i \times v \times \frac{n_i F p_o}{RT} \times (\text{Electrode area})^{-1} \qquad (1)$$

where $x_i$ is the volume fraction of certain product determined by online GC referenced to calibration curves from the standard gas sample (Airgas), $v$ is the flow rate of 50.0 s.c.c.m., $n_i$ is the number of electrons involved, $p_o$ = 101.3 kPa, F is the Faradaic constant, T = 298 K, and R is the gas constant. The corresponding FE at each potential is calculated by

$$FE = \frac{j_i}{j_{total}} \times 100\% \qquad (2)$$

One-dimensional 1H NMR spectra were collected on a Bruker AVIII 500 MHz NMR spectrometer to quantify the liquid products. Typically, 500 μL of electrolyte after electrolysis were mixed with 100 μL of $D_2O$ (Sigma Aldrich, 99.9 at% D) and 0.03 μL dimethyl sulfoxide (Sigma Aldrich, 99.9%) as internal standard. $^{13}$C NMR spectra were acquired using a Bruker AVIII 500 MHz NMR spectrometer at room temperature, to check the purity of the as-prepared HCOOH solution. Typically, 500 μl of electrolyte after electrolysis was mixed with 100 μL of $D_2O$ (Sigma Aldrich, 99.9 at.% D).

## Data availability

The data that support the findings of this study are available from the corresponding authors upon reasonable request

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

## Acknowledgements

This work was supported by Rice University. H.W. acknowledges support from the Welch Foundation Research Grant. H.W. is a CIFAR Azrieli Global Scholar in the Bio-inspired Solar Energy Program. C.X. acknowledges support from a J. Evans Attwell-Welch postdoctoral fellowship provided by the Smalley-Curl Institute. L.F. acknowledges support from the China Scholarship Council (CSC) (201806320253) and 2018 Zhejiang University Academic Award for Outstanding Doctoral Candidates.

## Author contributions

L.F. and C.X. contributed equally. Y.L. and H.W. supervised the project. L.F., C.X., and P.Z. performed the experimental work. L.F., C.X., and H.W. wrote the manuscript with the support from all authors.

## Competing interests

The authors declare no competing interests.
