## [Peer Review File · Nature Communications]

Reviewers' Comments:

Reviewer #1:

Remarks to the Author:

Refereeing: Electrochemical CO₂ reduction to high-concentration pure formic acid solutions in an all-solid-state reactor

The authors present an advance on their previous work published in Nature Energy 471 4, 776-785 (2019), with increased current density and product weight %. Although the data rests heavily on their previous work, the advances reported are enough to justify publication in Nature Communications; a 100% product stream has yet to be reported for electrocatalytic formic acid production, which is often seen as a significant hurdle in the application of this technology. Provided that the following points are addressed I believe this work should be published:

- Many grammatical errors are present throughout the text, which should be corrected before publication. This is particularly true in the introduction.
- The discussion of grain boundaries and their influence on catalytic current density is one possible explanation for the increased activity, however the increased surface area alone is sufficient to justify the large increase in current. High surface area catalysts are particularly important when using membrane electrode assemblies to increase contact between catalyst surface and polymer electrolyte.
- The activity of the OD-BI in the PSE cell should be provided as a comparison to the data from nBuLi-Bi presented in Figure 3d. This is the context in which catalyst comparison of the activity is most important.
- Can the authors provide a comment on why 30 mA cm⁻² was chosen for their stability experiments? As stability becomes a key metric for the future of this field it would be more appropriate to explore the stability at higher current density. Even if the cell quickly destabilises at high rates, it would help set a benchmark for future studies and would provide more information concerning electrolyser stability. This stability data should be provided.
- The authors should be careful with their statement that formic acid may be a H₂ carrier when their reactor uses H₂ oxidation as the anodic reaction.
- Line 291 – incredible should be replaced with a less subjective word, such as unprecedented.

Reviewer #2:

Remarks to the Author:

The authors modify the solid-electrolyte approach for production of liquid fuel solutions which was published in 2019 (Nature Energy, 4, 776-785(2019)). These concerns would need to be overcome:

1. The authors couple CO₂RR with HOR. Since hydrogen is a valuable feedstock, with a higher value than that of formate, it is not clear to me that the approach has a pathway to making sense economically. Also, long term operation seems to have been lost in this work compared to the Nature Energy paper, i.e. there is a substantial increase in the ohmic resistance induced by the membrane dehydration. A detailed techno-economic analysis that considered together the performance metrics, and the gains and losses over the Nature Energy people, could inform whether the route in this work is a promising one.

2. Instead of using a DI water flow into SSE, in this work, N₂ gas is used to extract the the formic acid, and the authors claim formic acid concentrations of 100wt%. In this system, CO₂RR is performed at the cathode, making one HCOO⁻ (formate) and one OH⁻. And from there the HCOO⁻ and OH⁻ transport through the AEM to combine with the two H⁺ at the anode to produce 1 mol of HCOOH (formic acid) and 1 mol of H₂O. This implies that there will always be water present inside the SSE. The theoretical maximum concentration in this case is only 17 M (~68 wt%). Also, since humidified

CO₂ (since H₂O needs to be present for CO₂RR) is fed from cathode, the water will unavoidably crossover to the SSE by electroosmotic drag, which will dilute the formic acid and will prevent achieving 100% wt concentration.

3. The authors claim later on that this system is not completely practical because the membranes need to be hydrated sufficiently to be able to conduct ions as well as maintain their physical and mechanical integrity. If it is claimed that it is possible to obtain 100 wt% formic acid, how is it possible to maintain sufficient humidification for the membranes, specifically after several hours of operation. This claim comes to the point there is 0 water production and 0 consumption inside the cell, I also doubt how the entire reaction can be initiated since there is no ion transport.

4. All work needs replicates and error bars.

5. The 100 hours stability was achieved at a low 30 mA cm⁻², even though there were current density demonstrations at least 15-fold higher than this current density in the paper. Even at this low current, the cell experienced a substantial increase in its working potential (the cell voltage increases from 1.4 V to 2.3 V), due mainly to the membrane dehydration, which prevents ion transport and reduces the efficiency of CO₂RR (HER becomes dominant). These points raise questions about whether there is a true net advantage / practical advantage over Nature Energy, 4, 776-785(2019)) If the authors could show 100 hours of stability at high reaction rates with concentrated formic acid production (at least above 50 wt%), this would highlight more of an advance.

Reviewer #3:

Remarks to the Author:

This manuscript by Wang et al. reported the construction of an all-solid-state electrochemical CO₂RR system for the continuous formic acid production. Even though a similar all-solid-state reactor design was recently reported by the same authors, inert gas stream was used as the carrier under the current investigation instead of deionized water, ultimately yielding highly concentrated and pure formic acid solution. This demonstration is very interesting to people working in the CO₂RR community. Only a few technical questions as noted below:

1) The Bi catalyst is separated from the PSE via an AEM, which ideally blocks the H⁺ cross-over. So where was the proton source for CO₂ reduction to formate at the cathode, specially when dry N₂ was used as the gas carrier?

2) HOR was used as the anodic reaction instead of OER, which was interesting. Pt was known to catalyze HOR at almost zero overpotential. So if CO₂RR can initiate on Bi at -0.65 V vs. RHE at the three-electrode configuration (Figure 3a), one would expect the full cell can be turned on at ~0.6-0.7 V instead of 1.3-1.4 V as demonstrated in Figure 3d. Please justify the obvious discrepancy.

3) Since the full cell is capable of delivering large current density up to 800 mA/cm². Its stability is recommended to be assessed at at least 100 mA/cm² instead of only 30 mA/cm² in Figure 3f.

4) How conductive was the PSE? Any impedance data? Did its thickness matter and impact the cell performance?

5) Is it economical to replace the OER with HOR at the anode? Please provide a techno-economic analysis if possible. What is the overall energy conversion efficiency in the current system?

Response to reviewer's comments:

We thank the reviewers for their constructive comments which have helped us to greatly improve our research and the quality of our manuscript. We have now included additional analysis and experiments to fully address the reviewers' concerns and suggestions. Below, we have addressed the points raised by reviewers one by one.

REVIEWER COMMENTS

Reviewer #1

Refereeing: Electrochemical CO₂ reduction to high-concentration pure formic acid solutions in an all-solid-state reactor. The authors present an advance on their previous work published in Nature Energy 471 4, 776-785 (2019), with increased current density and product weight %. Although the data rests heavily on their previous work, the advances reported are enough to justify publication in Nature Communications; a 100% product stream has yet to be reported for electrocatalytic formic acid production, which is often seen as a significant hurdle in the application of this technology. Provided that the following points are addressed I believe this work should be published:

Response

We highly appreciate the reviewer's praise and support of our work, as well as the following important suggestions which greatly improved the quality of our manuscript.

Comment 1

Many grammatical errors are present throughout the text, which should be corrected before publication. This is particularly true in the introduction.

Response

We are sorry for the errors. We have now checked the grammatical errors and revised them in this version.

Comment 2

The discussion of grain boundaries and their influence on catalytic current density is one possible explanation for the increased activity, however the increased surface area alone is sufficient to justify the large increase in current. High surface area catalysts are particularly important when using membrane electrode assemblies to increase contact between catalyst surface and polymer electrolyte.

Response

Thank you for raising this important point. The formate partial current density was not only related to the overall current density, but also influenced by the formate Faradaic efficiency. As shown in Fig. 3b (copied below), the formate Faradaic efficiency of nBuLi-Bi was much higher than that of OD-Bi at all applied potentials. In addition, the onset potential for formate of nBuLi-Bi was lower than OD-Bi. These results shown that the intrinsic CO₂RR activity of nBuLi-Bi was better than OD-Bi due to the enriched grain boundaries. We agree with the reviewer's opinion that the increased surface area can also increase the current density. In summary, the higher activity of nBuLi-Bi should be contributed from the grain boundaries, as well as the increased surface area. We have included extra discussions for this important point in the revised manuscript on page 6 and 7.

Figure R1. The formate FEs of Bi₂O₃ and nBuLi-Bi catalysts in a flow cell set-up using 1.0 M KHCO₃ electrolyte.

Comment 3

The activity of the OD-BI in the PSE cell should be provided as a comparison to the data from nBuLi-Bi presented in Figure 3d. This is the context in which catalyst comparison of the activity is most important.

Response

We appreciate the reviewer for this good suggestion. We have compared the intrinsic CO₂-to-formate performance of nBuLi-Bi and OD-Bi in a three-electrode flow reactor with a reference electrode. We believe that a three-electrode configuration cell is more suitable for catalyst study compared with the two-electrode full cell. The two-electrode full cell performance will be influenced by the cell assemble, such as the contact between the solid electrolyte and membrane. In contrast, the three-electrode flow reactor performance only related to the catalytic activity. As presented in the manuscript (Fig. 3a-c, copied below), the three-electrode flow reactor performance clearly showed that the nBuLi-Bi catalyst delivers much enhanced CO₂-to-formate performance compared with OD-Bi one, originating from the enriched grain boundaries and increased surface area.

Figure R2. Electrochemical CO₂ reduction performance of Bi₂O₃ and nBuLi-Bi. **(a)** The I-V curves of Bi₂O₃ and nBuLi-Bi catalysts in a flow cell set-up using 1.0 M KHCO₃ electrolyte. **(b)** The corresponding formate FE under different potentials. The maximal formate FE of nBuLi-Bi is 97% at -0.77V vs. RHE. **(c)** The comparison of formate partial current between Bi₂O₃ and nBuLi-Bi.

Comment 4

Can the authors provide a comment on why 30 mA cm⁻² was chosen for their stability experiments? As stability becomes a key metric for the future of this field it would be more appropriate to explore the stability at higher current density. Even if the cell quickly destabilises at high rates, it would help set a benchmark for future studies and would provide more information concerning electrolyser stability. This stability data should be provided.

Response

We strongly agree with the reviewer that the long-term stability under commercial-level current density is the key metric for the future of CO₂ reduction. In the traditional flow cell, the gas diffusion electrode (GDL) will flood within several hours under high current density

like 100 mA cm⁻². This is a common issue which still hinders the commercialization of GDL-based flow reactor for CO₂RR technology. To solve this problem, an anion exchange membrane (AEM) was used here to separate the GDL from the water stream. At the same time, the *in-situ* generated formate can cross the membrane into the PSE layer driven by the electrical field. While the Bi-catalyst based GDL will not flood even when continuously operated for 100 hours (DOI: 10.1038/s41560-019-0451-x), we found that the AEM will gradually decay. This is due to the AEM could be partially neutralized by CO₂ gas (DOI: 10.1149/1.2981860). As a result, the ionic conductivity of the AEM will gradually decrease, and then leads to increased cell voltage. The current density also plays a critical role here for AEM gradually decay (DOI: 10.1016/j.jpowsour.2017.07.117). Higher concentration of hydroxide ions will be generated simultaneously during CO₂RR under increased current density (CO₂ + H₂O + 2e⁻ → HCOO⁻ + OH⁻). Then, the CO₂ permeation flux will be increased according to the following equation:

With the increased current density, the ohmic resistance and electrode overpotential will increase (10.1016/j.electacta.2012.10.105), leading to the degradation of cell performance.

Considering the abovementioned reasons, we choose 30 mA cm⁻² (142.5 mA cell current) to test our catalyst and cell stability, in which ~0.1 M pure formic acid solution could be continuously produced for 100 hours (Fig. 3f). Besides, we also tested the stability at 100 mA cm⁻². As shown in Supplementary Fig. 9 (copied below), our solid-state cell can also stably run for at least 10 hours with slight increase in cell voltage. We believe that the development of CO₂-tolerant AEM could further push the performance of our solid-electrolyte cell to a new level.

Figure R3. Stability test with a fixed current density of 100 mA cm^{-2} .

Comment 5

The authors should be careful with their statement that formic acid may be a H_2 carrier when their reactor uses H_2 oxidation as the anodic reaction.

Response

Thank you for this comment. Hydrogen is a very clean and valuable fuel. However, the storage and transportation of hydrogen for practical use still remains as an unsolved challenge now. On the other hand, formic acid, which is a liquid fuel at ambient pressure and temperature, is much easier to be stored and transported compared with gaseous hydrogen. It shows very high volumetric hydrogen density of 53 g H_2 per liter. While we used hydrogen as the feedstock for HCOOH production, we believe it is a proper way to store the gaseous hydrogen into liquid fuel ($\text{CO}_2 + \text{H}_2 \rightarrow \text{HCOOH}$), which could be reversibly released from formic acid ($\text{HCOOH} \rightarrow \text{CO}_2 + \text{H}_2$). We have revised this point in the conclusion on page 13 to make it clearer.

Comment 6

Line 291 – incredible should be replaced with a less subjective word, such as unprecedented.

Response

Thanks for your good suggestion. We have replaced the word of “incredible” as unprecedented.

Reviewer #2

The authors modify the solid-electrolyte approach for production of liquid fuel solutions which was published in 2019 (Nature Energy, 4, 776-785(2019)). These concerns would need to be overcome:

Response

We appreciate the reviewer's constructive comments. Compared with our previous report, here we have developed a more efficient CO₂-to-formate catalyst, namely nBuLi-Bi, and an all-solid-state reactor to achieve pure HCOOH solution with much higher purity and concentration. Specifically, we proposed a facial and general method to fabricate grain boundary-enriched Bi-catalyst directly from commercial Bi₂O₃. Compared with our previous layered 2D-Bi catalyst where the layer-by-layer stacking could block the CO₂ diffusion pathway (DOI: 10.1038/s41560-019-0451-x), the porous nanoparticulate morphology of nBuLi-Bi now allows for more efficient CO₂ diffusions from GDL to the local active sites, and thus leading to much improved activity. Our nBuLi-Bi catalyst could deliver a maximal formate partial current of **~450 mA cm⁻²** whereas that of 2D-Bi is **~200 mA cm⁻²**. Furthermore, by reactor optimization we also achieved an almost 100% pure HCOOH for the first time. We have also addressed all comments raised by the reviewer, which has greatly improved the depth and rigor of this work.

Comment 1

The authors couple CO₂RR with HOR. Since hydrogen is a valuable feedstock, with a higher value than that of formate, it is not clear to me that the approach has a pathway to making sense economically. Also, long term operation seems to have been lost in this work compared to the Nature Energy paper, i.e. there is a substantial increase in the ohmic resistance induced by the membrane dehydration. A detailed techno-economic analysis that considered together the performance metrics, and the gains and losses over the Nature Energy people, could inform whether the route in this work is a promising one.

Response

We thank the reviewer for this valuable suggestion.

Electrochemical hydrogen evolution reaction has been gradually implemented in industry for mass H₂ generation. As the price of renewable electricity decreases, the cost of H₂ will drop further. However, storage and transportation of H₂ are still a remaining challenge. Compared with gas-phase H₂, formic acid, which is a liquid fuel under ambient conditions, is much easier to be stored and transported. It shows very high volumetric hydrogen density of 53g H₂ per liter. Thus, it is believed to be a good alternative H₂ carrier for practical use. While we used hydrogen as the feedstock for HCOOH production ($\text{CO}_2 + \text{H}_2 \rightarrow \text{HCOOH}$), it could be reversibly released from formic acid ($\text{HCOOH} \rightarrow \text{CO}_2 + \text{H}_2$). On the other hand, the as-prepared ultrapure HCOOH could be directly used as platform feedstock to synthesize other high-value-added C₂₊ chemicals (DOI: 10.1002/qua.22386). We have discussed this point in the revised manuscript on page 13.

Long term stability is still a key challenge in electrochemical CO₂ reduction. Different systems have different limitations. For example, traditional gas-diffusion-electrode (GDE) based flow cell suffers from flooding issue, the lifespan of the GDE-based reactor is only within few hours. We try to solve the flooding issue of the GDE by using an anion exchange membrane to separate the water stream and the catalyst electrode. As a result, we demonstrated a 100h stability of our nBuLi-Bi catalyst, which shows much improved lifespan compared with traditional flow reactor.

For two-electrode full cell, in our previous work, we used 0.5 M sulfuric acid for acidic oxygen evolution reaction at the anode to provide protons. However, sulfuric acid is a strong acid which is dangerous for practical application. Thus, hydrogen oxidation reaction was used to replace the acidic oxygen evolution reaction at anode to build-up a liquid-stream free reactor for safety. However, probably due to the imperfect contact between the Pt/C gas diffusion layer and the Nafion membrane, the voltage slightly increased after 50 hours. Of note, the HCOOH FE remains nearly constant, demonstrating the intrinsic good stability of the catalyst. Further work should focus on the electrode design to improve the ionic transportation between the gas diffusion layer and the membrane in order to further push forward the stability of the reactor.

Based on the reviewer's suggestion, we also compared the net cost of this work and our previous work according to the stability test data as following.

CO₂//SE//H₂ 4.75 cm² cell (this work): Operation condition: 1.9 V (average voltage among 100 hours stability test); 142.5 mA; Average HCOOH Faradaic efficiency: 80%; Production rate: 0.45 mmol cm⁻² h⁻¹ (0.083 g h⁻¹); Operation time: 100 hours. **Generated HCOOH: 9.8g.** Consumed CO₂: 9.37 g. Consumed H₂: 0.43 g. Electricity cost: 3 cents/kWh × 1.9V × 0.1425 A × 100 h = 81 cents. HOR catalyst cost: \$75/g × 0.7mg/cm² × 4.75cm² = 24.9 cents. CO₂ cost: \$0.03/kg × 7.94g = 0.024 cents. H₂ cost: \$3.9/kg × 0.43g = 0.167 cents. Water cost: \$0.008/kg × 16mL/h × 100h × 1g/mL = 1.3 cents. **Total cost: 107 cents.**

CO₂//SE//H₂O cell (Previous work): Operation condition: 3.0 V (120 mA); Average HCOOH Faradaic efficiency: 80%; Production rate: 0.45 mmol cm⁻² h⁻¹ (0.083 g h⁻¹); Operation time: 100 hours. **Generated HCOOH: 8.3g.** Consumed CO₂: 7.94 g. Electricity cost: 3 cents/kWh × 3.0V × 0.12 A × 100 h = 108 cents. OER catalyst cost: \$197/g × 0.5mg/cm² × 4cm² = 39.4 cents. CO₂ cost: \$0.03/kg × 7.94g = 0.024 cents. Water cost: \$0.008/kg × 16.2 mL/h × 100h × 1g/mL = 1.3 cents. **Total cost: 149 cents.**

(Note: Both of these two designs need same membranes and bismuth catalysts, so we did not include these costs in the estimation)

Data sources:

CO₂ and H₂ price: DOI: 10.1126/science.aav3506

H₂O price: The industrial water in Texas is \$1.91 per 1000 gallons (<https://www.fbgtx.org/673/IndustrialWater-Rates>). Only 1 to 3 cents are needed to deionize one gallon of water (<https://blog.uswatersystems.com/2012/08/de-ionization-101/>). Thus, the price of DI water is estimated to be ~ 3 cents/gallon or \$0.008/kg.

10% Ir/C and 10% Pt/C price: Fuel cell store

According to the above-mentioned preliminary comparison, we believe that the upgraded design combined with the new Bi catalyst in this work is much more promising for practical applications. The related discussion has been added into the revised supplementary materials as supplementary note 1 and copied below.

Note 1: Preliminary estimation of the production cost for the electrosynthesis of HCOOH using the 4.75 cm² CO₂//SE//H₂ cell. We only calculated the costs of energy and feedstock input; no other costs associated with practical production or infrastructure were included. The market price of formic acid is \$0.74/kg (DOI: 10.1021/acs.iecr.7b03514).

CO₂//SE//H₂ cell: Operation condition: 1.98 V (3000 mA); Production rate: 7.55 mmol cm⁻² h⁻¹ (1.65 g h⁻¹)

$$m_{\text{HCOOH}} = \frac{1 \text{ kWh}}{1.98 \text{ V} \times 3 \text{ A}} \times 1.65 \text{ g h}^{-1} = \sim 277.8 \text{ g}$$

Thus, we can obtain 0.2778 kg HCOOH using 1 kWh electricity. This 0.2778 kg HCOOH consumes 0.2657 kg CO₂ and 0.012 kg H₂, where the industrial CO₂ price is \$0.03/kg, and the price of H₂ is ~ \$3.9/kg (DOI: 10.1126/science.aav3506). Therefore, the CO₂ cost is 0.8 cents, H₂ cost is 4.7 cents. Assuming the price of electricity is 3 cents/kWh, we can roughly estimate a HCOOH production cost of ca. \$0.31/kg-HCOOH without considering the cost of DI water. The industrial water in Texas is \$1.91 per 1000 gallon (<https://www.fbgtx.org/673/IndustrialWater-Rates>). Only 1 to 3 cents are needed to deionize one gallon of water (<https://blog.uswatersystems.com/2012/08/de-ionization-101/>). Thus, the price of DI water is estimated to be ~ 3 cents/gallon or \$0.008/kg, which only adds a marginal cost to the HCOOH production cost of \$0.31/kg.

Comment 2

Instead of using a DI water flow into SSE, in this work, N₂ gas is used to extract the the formic acid, and the authors claim formic acid concentrations of 100wt%. In this system, CO₂RR is performed at the cathode, making one HCOO⁻ (formate) and one OH⁻. And from there the HCOO⁻ and OH⁻ transport through the AEM to combine with the two H⁺ at the anode to produce 1 mol of HCOOH (formic acid) and 1 mol of H₂O. This implies that there will always be water present inside the SSE. The theoretical maximum concentration in this case is only 17 M (~68 wt%). Also, since humidified CO₂ (since H₂O needs to be present for CO₂RR) is fed from cathode, the water will unavoidably crossover to the SSE by electroosmotic drag, which will dilute the formic acid and will prevent achieving 100% wt concentration.

Response

Thank you for raising this important point. Electrosynthesis of HCOOH in this case can be decoupled into two half-cell reactions (Eqs. 1 and 2), followed by the ionic recombination process (Eq. 3):

So, in principle, there is no need for extra water. As we have mentioned in the experimental section that **dry** CO₂, N₂, and H₂ were used for pure HCOOH vapour (almost 100%) production to avoid the introduction of any water vapor. In this system, CO₂RR is performed at the cathode, making one HCOO⁻ (formate) and one OH⁻, and simultaneously consuming one water (equation 1). *Since only dry CO₂ is provided, the surface water of the AEM membrane will involve in the electrochemical CO₂ reduction process to initiate the CO₂-to-formate conversion* (DOI: 10.1021/acs.macromol.8b00034). Next, the generated HCOO⁻ and OH⁻ will transport through the AEM into the porous PSE layer to combine with the two H⁺ to produce one HCOOH (formic acid) and one H₂O. This recombination process will occur at the AEM/PSE interface since a solid proton conductor was used in our case (DOI: 10.1038/s41560-019-0451-x). Then, the recombined H₂O will be immediately absorbed by the surface imidazole groups of the AEM membrane (DOI: 10.1016/j.cplett.2004.04.065). Thus, no net water was generated in the entire electrochemical CO₂ reduction cycle. What is more, while trace amount of water vapor will be introduced into the system from environmental moisture, the AEM, CEM and porous PSE is superhydrophilic, which can absorb the water molecules. Moreover, compared with water, formic acid is much more volatile, which makes it easier to be released by the nitrogen flow. All the features, taken together, make the production of ultrapure HCOOH vapor possible. This process has been illustrated in supplementary Fig. 14 and copied below. This related discussion has been added into revised supplementary information as Supplementary note 2.

Figure R4. Schematic illustration of water consumption and generation in the AEM.

Comment 3

The authors claim later on that this system is not completely practical because the membranes need to be hydrated sufficiently to be able to conduct ions as well as maintain their physical and mechanical integrity. If it is claimed that it is possible obtain 100 wt% formic acid, how it is possible to maintain sufficient humidification for the membranes, specifically after several hours of operation. This claim comes to the point there is 0 water production and 0 consumption inside the cell, I also doubt how the entire reaction can be initiated since there is no ion transport.

Response

In our reactor, electrosynthesis of HCOOH can be decoupled into two half-cell reactions (Eqs. 1 and 2), followed by the ionic recombination process (Eq. 3):

So, HCOO^- , OH^- and protons are the charge carriers in this case, which are transported within the reactor. In principle, according to above reactions, there is 0 water production and 0 consumption inside the cell. This process is clearly illustrated in scheme in the comment 2.

Initially, the fresh membranes are highly hydrated. The catalyst-coated gas diffusion layer was closely contact with the membrane, the surface water of the membrane will involve in the electrochemical CO_2 reduction (DOI: 10.1021/acs.macromol.8b00034) and initiate the entire reaction. No extra water is required for the CO_2 reduction process. To prepare ~100% pure HCOOH vapor, the dry gas flow was used in our study. However, the dry gas flows (N_2 , CO_2 and H_2) will gradually and slightly decrease the surface water of the membranes. The dehydrated membranes after long-term test will induce the decay of the performance of the cell, due to the limited ionic conductivity of the dry membranes. As a result, the generation of almost 100% pure HCOOH can only be continuously operated for several hours. However, when humidified feedstock gases (H_2 and CO_2) were provided and DI water was used to release the HCOOH , the reactor can be continuously operated for at least 100 hours (Fig. 3f and copied below). Now, we are trying to make hydrogel-like membranes, which can tightly lock in moisture, to overcome this disadvantage of the newly proposed cell. We believe that the advances of membrane technology could significantly extend the stability of such a design.

Figure R5. Stability test of continuous generation of pure formic acid solutions with concentrations over 0.1 M. The cell current and DI water flow-rate was fixed at 30 mA cm^{-2} and 16 mL h^{-1} , respectively. The full cell voltage showed a slight increase after 50 hours, which could stem from the anode/CEM interface.

Comment 4

All work needs replicates and error bars.

Response

We highly appreciate the reviewer for this important suggestion. Now, the error bars have been included for revised Fig. 3, Fig. 4, and copied below.

Figure R6. Electrochemical CO₂ reduction performance of Bi₂O₃ and nBuLi-Bi. **(a)** The I-V curves of Bi₂O₃ and nBuLi-Bi catalysts in a flow cell set-up using 1.0 M KHCO₃ electrolyte. **(b)** The corresponding formate FE under different potentials. The maximal formate FE of nBuLi-Bi is 97% at -0.77V vs. RHE. **(c)** The comparison of formate partial current between Bi₂O₃ and nBuLi-Bi. The maximal formate partial current on nBuLi-Bi reached to 450 mA cm⁻², which represents a four-fold improvement from Bi₂O₃. **(d)** The I-V curve of PSE cell with nBuLi-Bi and Pt/C as the CO₂RR and HOR catalysts, respectively, where DI water was used to release the produced formic acid molecules. **(e)** The corresponding FE of different products under different cell voltages. **(f)** Stability test of continuous generation of pure formic acid solutions with concentrations over 0.1 M. The cell current and DI water flow-rate was fixed at 30 mA cm⁻² and 16 mL h⁻¹, respectively. The full cell voltage showed a slight increase after 50 hours, which could stem from the anode/CEM interface. The error bars represent two independent tests.

Figure R7. All-solid-state electrochemical CO₂RR reactor for high concentration formic acid vapor. **(a)** The I-V curve of all-solid-state reactor using 100 sccm humidified N₂. **(b)** The corresponding FE of formic acid vapor under different cell voltages. **(c)** Performance comparison between DI water flow and humidified N₂ flow. Due to the high formic acid vapor release rate, humidified N₂ flow can maintain a high product selectivity while achieving concentrated formic acids. **(d)** The dependence of formic acid product concentration on the N₂ gas flow rate under a fixed overall current density of 200 mA cm⁻². By utilizing dry N₂ gas flow, 100 wt.% pure formic acid was achieved. **(e)** The NMR spectrum of formic acid product obtained via dry N₂ flow, compared with that of commercial formic acid product with a marked purity of > 96 wt.%. The normalized integral area of our formic acid products was even slightly higher than commercial product. **(f)** Stability test of CO₂RR to formic acid vapor under a current density of 30 mA cm⁻². The error bars represent three independent tests.

Comment 5

The 100 hours stability was achieved at a low 30 mA cm⁻², even though there were current density demonstrations at least 15-fold higher than this current density in the paper. Even at this low current, the cell experienced a substantial increase in its working potential (the cell voltage increases from 1.4 V to 2.3 V), due mainly to the membrane dehydration, which prevents ion transport and reduces the efficiency of CO₂RR (HER becomes dominant). These points raise questions about whether there is a true net advantage / practical advantage over Nature Energy, 4, 776-785(2019)). If the authors could show 100 hours of stability at high reaction rates with concentrated formic acid production (at least above 50 wt%), this would highlight more of an advance.

Response

Thank you for raising this important point. In this study, we use anion exchange membrane to solve the flooding issue for GDEs. However, there are still some unsolved challenges using anion exchange membrane in CO₂ reduction. It is well known that the alkaline fuel cell system is sensitive to CO₂ gas due to the alkali could be neutralized by CO₂ gas (DOI: 10.1149/1.2981860,). As a result, the ionic conductivity will decrease, and thus leading to increased cell voltage. For CO₂ reduction, the AEM was exposed to pure CO₂ atmosphere, such a high concentration CO₂ will continuously degrade the anion exchange membrane. Besides, the current density has actually a significant effect for AEM under CO₂ atmosphere (DOI: 10.1016/j.jpowsour.2017.07.117). With the increased current density, the ohmic resistance and electrode overpotential will increase (DOI: 10.1016/j.electacta.2012.10.105), leading to the degradation of cell performance.

Considering the abovementioned reason, we choose 30 mA cm⁻² to test our catalyst and reactor stability. The total cell current is 142.5 mA, which can continuously generate ~0.1 M pure formic acid solution for 100 h. Besides, as shown in Fig. 5F, we can continuously generate ~0.35 M pure formic acid for 30 hours, which is better than our previous work. Besides, we also tested the stability at 100 mA cm⁻². As shown in Supplementary Fig. 9 and copied bellow, our solid-state cell can stable run for 10 hours without obvious degradation. Further work should focus on the optimization of membrane design to further improve the long-term stability for practical application.

Figure R8. Stability test with a fixed current density of 100 mA cm⁻².

Reviewer #3

This manuscript by Wang et al. reported the construction of an all-solid-state electrochemical CO₂RR system for the continuous formic acid production. Even though a similar all-solid-state reactor design was recently reported by the same authors, inert gas stream was used as the carrier under the current investigation instead of deionized water, ultimately yielding highly concentrated and pure formic acid solution. This demonstration is very interesting to people working in the CO₂RR community. Only a few technical questions as noted below:

Response

We appreciate the reviewer's support of our work for publication, as well as the suggestions which have substantially improved the quality of our manuscript. We have addressed all the questions raised by the reviewer.

Comment 1

The Bi catalyst is separated from the PSE via an AEM, which ideally blocks the H⁺ cross-over. So where was the proton source for CO₂ reduction to formate at the cathode, specially when dry N₂ was used as the gas carrier?

Response

In our study, electrosynthesis of HCOOH can be decoupled into two half-cell reactions (Eqs. 1 and 2), followed by the ionic recombination process (Eq. 3):

In principle, according to above reactions, there is 0 water production and 0 consumption inside the cell. This process is illustrated in below scheme. Initially, the fresh membranes are highly hydrated. The catalyst-coated gas diffusion layer was closely contact with the membrane, the surface water of the membrane will involve in the electrochemical CO₂

reduction (DOI: 10.1021/acs.macromol.8b00034) and initiate the entire reaction. This related discussion has been added into revised supplementary information as Supplementary note 2.

Figure R9. Schematic illustration of water consumption and generation in the AEM.

Comment 2

HOR was used as the anodic reaction instead of OER, which was interesting. Pt was known to catalyze HOR at almost zero overpotential. So if CO₂RR can initiate on Bi at -0.65 V vs. RHE at the three-electrode configuration (Figure 3a), one would expect the full cell can be turned on at ~0.6-0.7 V instead of 1.3-1.4 V as demonstrated in Figure 3d. Please justify the obvious discrepancy.

Response

Thank you for this good point. Ideally, a 0.6-0.7 V cell voltage is enough to drive the whole reaction. But extra voltage is required to overcome the charge transfer resistance between the porous solid-state electrolyte and membranes (DOI: 10.1016/j.cattod.2005.03.074), as well as the resistance from the imperfect contact between the catalyst and membranes. As a result, the full cell voltage was slightly higher than theoretical estimation.

Comment 3

Since the full cell is capable of delivering large current density up to 800 mA/cm². Its stability is recommended to be assessed at at least 100 mA/cm² instead of only 30 mA/cm² in

Response

Thank you for raising this important point. In this study, we use anion exchange membrane to solve the flooding issues. However, there are still some unsolved challenges using anion exchange membrane in CO₂ reduction. It is well known that the alkaline fuel cell system is extremely sensitive to CO₂ gas because the alkali should be neutralized by CO₂ gas (DOI: 10.1149/1.2981860). As a result, the ionic conductivity will decrease and lead to increased cell voltage. For CO₂ reduction, the anion exchange was exposed to pure CO₂ atmosphere, such high concentration CO₂ will continuously degrade the anion exchange membrane. Besides, the current density has actually a significant effect for AEM under CO₂ atmosphere (DOI: 10.1016/j.jpowsour.2017.07.117). With the increased current density, the ohmic resistance and electrode overpotential will increase (DOI: 10.1016/j.electacta.2012.10.105), leading to the degradation of cell performance.

Considering the abovementioned reason, we choose 30 mA cm⁻² to test our catalyst and cell stability. The total cell current is 142.5 mA, which can continuously generate ~0.1 M pure formic acid solution for 100 h. Besides, we also tested the stability at 100 mA cm⁻². As shown in Supplementary Fig. 9, our solid-state cell can stable run for 10 hours without obvious degradation. Future work should focus on the optimization of anion exchange membrane design.

Figure R10. Stability test with a fixed current density of 100 mA cm⁻².

Comment 4

How conductive was the PSE? Any impedance data? Did its thickness matter and impact the

cell performance?

Response

In our previous work (DOI: 10.1038/s41560-019-0451-x), we have investigated and optimized the conductivity of the PSE in detail.

Figure R11. Electrochemical impedance spectra for the solid-state proton conductors. The impedance measurement of SSE-50 (~50 μm particle size) and SSE-300 (~300 μm particle size) solid proton ion conductors using an asymmetric cell with two stainless steel electrodes. The ionic conductivity of the solid electrolyte can be estimated from the resistivity (R) measured using the following equation: $R = \rho \frac{l}{A}$; where R represents the resistance of the electrolyte, ρ is resistivity, l is the distance between two electrodes, and A equals the cross-sectional area. The ionic conductivity κ of the electrolyte is, therefore, the reciprocal of the resistivity (ρ). The calculated ionic conductivity of the highly porous SSE-50 electrolyte is ca. 0.018 S cm^{-1} whereas that of porous SSE-300 solid electrolyte is ca. 0.0064 S cm^{-1} . The estimated charge transfer resistivity of SSE-50 and SSE-300 solid proton conductor is 12.5 and $24.5 \text{ } \Omega \text{ cm}^2$, respectively.

According to our previous work, we use SSE-50 in this work for fast ion-conducting. We found that the thickness of the PSE layer will affect the overall performance of cell, since thicker PSE layer will cause higher ohmic loss. Thus, in our case a 3 mm thick PSE layer was used, which can efficiently collect the formed HCOOH and also leads to acceptable resistance of 2~3 ohm.

Comment 5

Is it economical to replace the OER with HOR at the anode? Please provide an techno-economic analysis if possible. What is the overall energy conversion efficiency in the current system?

Response

Thank you for raising this important point. The HOR is used to replace the OER in our case in order to avoid the use of liquid electrolyte in the anode. This strategy can make it possible for generation of ~100 wt.% pure formic acid. We have estimated the techno-economic analysis in the revised supplementary materials as Supplementary note 1 and copied below.

Note 1: Preliminary estimation of the production cost for the electrosynthesis of HCOOH using the 4.75 cm² CO₂//SE//H₂ cell. We only calculated the costs of energy and feedstock input; no other costs associated with practical production or infrastructure were included. The market price of formic acid is \$0.74/kg (DOI: 10.1021/acs.iecr.7b03514).

CO₂//SE//H₂ cell: Operation condition: 1.98 V (3000 mA); Production rate: 7.55 mmol cm⁻² h⁻¹ (1.65 g h⁻¹)

$$m_{\text{HCOOH}} = \frac{1 \text{ kWh}}{1.98 \text{ V} \times 3 \text{ A}} \times 1.65 \text{ g h}^{-1} = \sim 277.8 \text{ g}$$

Thus, we can obtain 0.2778 kg HCOOH using 1 kWh electricity. This 0.2778 kg HCOOH consumes 0.2657 kg CO₂ and 0.012 kg H₂, where the industrial CO₂ price is \$0.03/kg, and the price of H₂ is ~ \$3.9/kg (DOI: 10.1126/science.aav3506). Therefore, the CO₂ cost is 0.8 cents, H₂ cost is 4.7 cents. Assuming the price of electricity is 3 cents/kWh, we can roughly estimate a HCOOH production cost of ca. \$0.31/kg-HCOOH without considering the cost of DI water. The industrial water in Texas is \$1.91 per 1000 gallon (<https://www.fbgtx.org/673/IndustrialWater-Rates>). Only 1 to 3 cents are needed to deionize one gallon of water (<https://blog.uswatersystems.com/2012/08/de-ionization-101/>). Thus, the price of DI water is estimated to be ~ 3 cents/gallon or \$0.008/kg, which only adds a marginal cost to the HCOOH production cost of \$0.31/kg.

The energy efficiency of CO₂RR is calculated using the following equations when oxygen evolution reaction was used in the anode:

$$\text{Energy efficiency} = \frac{E_{\text{OER}} - E_{\text{CO}_2\text{-to-HCOOH}}}{E_{\text{cell}}} F E_{\text{HCOOH}}$$

where E_{OER} (1.23V versus RHE) and $E_{\text{CO}_2\text{-to-HCOOH}}$ (-0.17 versus RHE) are the theoretical potential for OER and CO_2 reduction to HCOOH, respectively. E_{cell} is the required cell voltage.

However, in this study, we cannot use this concept for energy efficiency calculation since H_2 fuel was used in the anode as a feedstock. If we neglect the consumption of H_2 fuels, the electricity-to-HCOOH efficiency was ~10%.

Reviewers' Comments:

Reviewer #1:

Remarks to the Author:

The authors have done a good job in providing an improved manuscript.

To publish I would still like to see a comparison of the OD-BI in the PSE cell to the data from nBuLi-Bi presented in Figure 3d. As the authors say in their response letter, the formation of a polymer-based electrolyte reactor has many complications, therefore this comparison would be really useful to demonstrate such complications to the reader, particularly as this group are really ahead of the field in this regard. Even if nBuLi-Bi and OD-BI aren't too different in the PSE cell, this data would be hugely interesting as the field tries to establish which aspects of reactor design are most important.

It would also be great for the authors to briefly mention that they were prevented from undertaking stability studies at higher current density due to concerns with the AEM - this is obviously out of the author's control, so won't reflect badly on the manuscript. The reason I feel this information is important is that it may help inspire others to develop more suitable and robust AEMs in future.

Great work! I hope these comments are helpful.

Reviewer #2:

Remarks to the Author:

Two of the three referees feel that this is a sufficient advance over the prior related Wang group work, and I am happy to go along with the majority view. The senior corresponding author needs to edit the manuscript for grammar, diction, and rigorous scientific style.

Reviewer #3:

Remarks to the Author:

I am satisfied with the revisions.

Response to reviewers' comments:

We thank the editor and reviewers for their constructive comments which have helped us to greatly improve our research and the quality of our manuscript. Below, we address the points raised by reviewers one by one.

Reviewer #1

The authors have done a good job in providing an improved manuscript.

Response

We thank the reviewer's positive response.

Comment 1

To publish I would still like to see a comparison of the OD-BI in the PSE cell to the data from nBuLi-Bi presented in Figure 3d. As the authors say in their response letter, the formation of a polymer-based electrolyte reactor has many complications, therefore this comparison would be really useful to demonstrate such complications to the reader, particularly as this group are really ahead of the field in this regard. Even if nBuLi-Bi and OD-BI aren't too different in the PSE cell, this data would be hugely interesting as the field tries to establish which aspects of reactor design are most important.

Response

Thank you for raising this important point. We have tested the performance of the OD-Bi in the PSE cell, as shown in Supplementary Fig. 8 (copied below). Compared with nBuLi-Bi, OD-Bi required much higher cell voltage to deliver the same current density in the PSE cell. The maximum formic acid partial current density was only 174.4 mA cm^{-2} at a cell voltage of 2.44 V.

Figure R1. CO₂RR performance of OD-Bi using PSE reactor. a, The I-V curve of OD-Bi//PSE//Pt/C cell, where DI water was used to release the produced formic acid molecules. **b,** The corresponding FEs of different products under different cell voltages.

Comment 2

It would also be great for the authors to briefly mention that they were prevented from undertaking stability studies at higher current density due to concerns with the AEM - this is obviously out of the author's control, so won't reflect badly on the manuscript. The reason I feel this information is important is that it may help inspire others to develop more suitable and robust AEMs in future.

Response

Thanks for your good suggestion. We have discussed this important point on page 10 in the revised manuscript (copied below).

The quick degradation of CO₂RR stability at high current density could be mainly caused by the degradation of AEM, since the AEM could be partially neutralized by CO₂ gas. As a result, the ionic conductivity of the AEM will gradually decrease, and thus leads to increased cell voltage. When the current density increased, higher concentration of hydroxide ions will be generated simultaneously during CO₂RR. Then, the CO₂ permeation flux will increase, leading to faster CO₂RR performance decay.

Comment 3

Great work! I hope these comments are helpful.

Response

Thank you for your helpful suggestions which have helped us to greatly improve our research and the quality of our manuscript.

Reviewer #2

Two of the three referees feel that this is a sufficient advance over the prior related Wang group work, and I am happy to go along with the majority view. The senior corresponding author needs to edit the manuscript for grammar, diction, and rigorous scientific style.

Response

We thank the reviewer's positive response. We have carefully checked the grammar, diction, and scientific style and corrected these mistakes.

Reviewer #3

I am satisfied with the revisions.

Response

We thank the reviewer's positive response.